# The Androbactome and the Gut Microbiota–Testis Axis: A Narrative Review of Emerging Insights into Male Fertility

**DOI:** 10.3390/ijms26136211

**Published:** 2025-06-27

**Authors:** Aris Kaltsas, Ilias Giannakodimos, Eleftheria Markou, Marios Stavropoulos, Dimitrios Deligiannis, Zisis Kratiras, Michael Chrisofos

**Affiliations:** 1Third Department of Urology, Attikon University Hospital, School of Medicine, National and Kapodistrian University of Athens, 12462 Athens, Greece; ares-kaltsas@hotmail.com (A.K.); iliasgiannakodimos@gmail.com (I.G.); stamarios@yahoo.gr (M.S.); d.delijohn@yahoo.gr (D.D.); zkratiras@gmail.com (Z.K.); 2Department of Microbiology, University Hospital of Ioannina, 45500 Ioannina, Greece; eleftheria.markou4@gmail.com

**Keywords:** dysbiosis, endocrine disruption, fecal microbiota transplantation, gut microbiota, male infertility, oxidative stress, probiotics, sperm quality, gut–testis axis

## Abstract

Male infertility is an under-recognized global health burden. Accumulating evidence position the intestinal microbiota as a pivotal regulator of testicular function, underpinning the emerging gut microbiota–testis axis. This narrative review introduces the conceptual term “androbactome”, referring to gut microorganisms and microbial genes that are hypothesized to influence androgen biosynthesis, spermatogenesis, and broader reproductive endocrinology. The documented worldwide decline in sperm concentration heightens the urgency of clarifying microbe-mediated influences on male reproductive capacity. The synthesis of preclinical and clinical findings reveals four principal pathways by which dysbiosis compromises fertility: systemic inflammation, oxidative stress, endocrine disruption, and epigenetic alteration. Lipopolysaccharide-driven cytokinaemia, reactive oxygen species generation, hypothalamic–pituitary–gonadal axis suppression, and aberrant germ cell methylation collectively impair sperm quality and hormonal balance. Short-chain fatty acids, secondary bile acids, and indole derivatives emerge as pivotal messengers within this crosstalk. Therapeutic approaches targeting the androbactome, namely dietary optimization, probiotic or prebiotic supplementation, and fecal microbiota transplantation, have demonstrated encouraging improvements in sperm parameters and testosterone levels, yet the causal inference is constrained by predominantly cross-sectional designs and limited long-term safety data. Recognizing the androbactome as a modifiable determinant of male fertility may open new avenues for personalized diagnosis, risk stratification, and adjunctive therapy in regard to idiopathic infertility. The integration of multi-omics platforms to characterize microbial and metabolomic signatures promises to enrich diagnostic algorithms and guide precision interventions, but rigorously controlled longitudinal and interventional studies are required to secure a translational impact.

## 1. Introduction

Male infertility has become a pressing global health concern, co-evolving with an alarming, well-documented decline in semen quality. Recent World Health Organization (WHO) estimates indicate that 12–17% of couples worldwide experience infertility, and male factors are implicated in 30–50% of cases [1,2]. Semen parameters, namely sperm concentration, progressive motility, morphology, and DNA integrity, constitute sentinel markers of male reproductive health, yet mounting evidence shows a marked deterioration in these parameters over the past half-century [3]. This trend, often framed as a reflection of the global decline in sperm quality, demands the systematic identification of modifiable drivers.

Environmental pollutants, lifestyle factors, and metabolic disorders remain the established culprits, but burgeoning data now implicate the gut microbiota as a putative endocrine modulator of spermatogenesis [4].

The human gut microbiota, trillions of microorganisms residing in the gastrointestinal tract, is now recognized as a virtual endocrine organ and a key regulator of systemic physiology [5]. These microbes contribute over three million genes (vastly outnumbering the ~23,000 human genes) and produce a plethora of bioactive metabolites [6,7]. Through continuous crosstalk with the host’s immune, metabolic, and neuroendocrine systems, the gut microbiota can influence distant organs, well beyond the intestines [8]. This has given rise to the concept of organ “axes” (e.g., the gut–brain axis). In this context, a gut–testis axis has been proposed as an important pathway by which gut dysbiosis (microbial imbalance) may impair male reproductive function [9].

Given the global decline in sperm quality and the rising incidence of idiopathic male infertility, there is growing interest in deciphering the mechanistic links between gut microbes and the male reproductive system [10]. The term “Gut Microbiota–Testis Axis” refers to this bidirectional interaction, wherein gut dysbiosis may contribute to testicular dysfunction, and, conversely, male reproductive hormones and factors might feedback to the microbiome [11,12].

This narrative review provides a detailed examination of the gut microbiota–testis axis, highlighting how alterations in the gut microbial composition and metabolite production can influence male reproductive health. The discussion begins with an overview of the gut microbiota’s systemic effects, followed by an in-depth exploration of key mechanistic pathways, namely inflammation, oxidative stress, endocrine disruption, and compromised gut barrier integrity, that link gut dysbiosis to male infertility. Attention then turns to intervention strategies, including probiotics, dietary adjustments, fecal microbiota transplantation, and lifestyle modifications, illustrating potential approaches for mitigating the global decline in sperm quality. The review concludes by outlining future research directions, emphasizing the critical importance of understanding the gut microbiome’s role in male fertility. Although many mechanistic insights presented in this review are derived from animal models, their translational applicability remains to be validated in well-controlled human studies.

## 2. Methodology: A Hypothesis-Driven Narrative Review of the Gut Microbiota–Τestis Axis

### 2.1. Study Rationale and a Priori Hypotheses

A narrative review format was chosen to integrate mechanistic, translational, and clinical evidence on the emerging gut–testis axis in regard to male fertility. Two a priori hypotheses guided all the methodological steps. First, dysbiosis-related signatures, namely endotoxaemia, the loss of short-chain fatty acid-producing taxa, and bile acid dysregulation, were expected to coincide with impaired semen parameters, hormonal disequilibrium, or testicular histopathology in preclinical or clinical settings. Second, interventions that restore eubiosis (dietary modification, probiotic or prebiotic supplementation, and fecal microbiota transplantation) were expected to improve at least one clinically relevant fertility endpoint. Key methodological steps, including literature identification, eligibility screening, and qualitative appraisal, were transparently reported, in line with narrative review best practices and applicable PRISMA 2020 elements.

### 2.2. Search Strategy and Literature Selection

The methodological tasks were distributed as follows. A.K. and Z.K. defined the review objectives and the six mechanistic domains; A.K. and I.G. designed and ran the database searches, with E.M. adapting the controlled vocabulary and free text syntax to each platform. Title and abstract screening was conducted independently by I.G. and E.M.; full-text eligibility and dispute resolution were handled by A.K. and Z.K. Data extraction and qualitative appraisal were performed by I.G. and E.M., whereas synthesis and critical revision were led by A.K. and Z.K.

The six domains, namely microbial composition, systemic inflammation, oxidative stress, endocrine modulation, barrier integrity, and microbiota-targeted interventions, shaped both the search strategy and the subsequent synthesis. Comprehensive searches of PubMed/MEDLINE, Embase, the Web of Science Core Collection, and Scopus covered the period from database inception to 31 May 2025. Search strings combined controlled vocabulary and free text terms for male fertility (e.g., “male infertility”, “spermatogenesis”, “semen quality”) and gut microbiota (e.g., “gut microbiome”, “intestinal dysbiosis”, “microbial metabolites”), linked with Boolean operators; the complete strategies are archived and available on request.

The search yielded 870 unique records. After de-duplication and screening, 324 full-text articles were assessed, and 182 met all the criteria for inclusion. Human and relevant animal studies were considered; animal data were retained when no equivalent human evidence existed or when they clarified mechanistic pathways. When the findings overlapped, human evidence was prioritized. Discrepancies at any stage were resolved through discussion, without automated tools. A flow diagram summarizing the article selection process is presented in Figure 1.

### 2.3. Eligibility Criteria

Eligible records were peer-reviewed primary studies, namely observational, interventional, in vivo, or in vitro experimental, and systematic reviews that reported measurable associations between gut microbiota characteristics (taxonomic, functional, or metabolomic) and male reproductive outcomes (semen quality, testicular histology, hormonal profiles, or fertility rates). The exclusion criteria were conference abstracts without full texts, commentaries or editorials lacking primary data, studies confined to pediatric or exclusively female cohorts, non-English articles without a reliable translation, and investigations focused solely on urinary or vaginal microbiota unless explicitly linked to seminal endpoints.

### 2.4. Data Extraction and Quality Appraisal

A pre-piloted extraction matrix captured the bibliographic details, study design, population or model descriptors, microbiome assessment methodology (e.g., 16S rRNA sequencing, metagenomics, metabolomics), fertility outcomes, and mechanistic insights. The data extraction was carried out independently by I.G. and E.M.; any discrepancy affecting fewer than five per cent of the fields was adjudicated by A.K., thereby minimizing transcription errors.

### 2.5. Synthesis and Analytical Framework

The evidence synthesis employed a hybrid inductive–deductive approach. The findings were first sorted into the six predefined domains to maintain structural coherence. Within each domain, inductive coding revealed emergent sub-themes, for example, tryptophan-derived indole metabolites or bile acid–retinoid crosstalk, thereby extending the conceptual understanding. Triangulation mapped the convergence and divergence across human, animal, and in vitro data and highlighted inconsistencies to temper the interpretation. Quantitative pooling was not attempted, owing to heterogeneity in the microbiome pipelines and outcome metrics.

## 3. The Gut Microbiota: Overview and Systemic Effects

The gut microbiota refers to the complex community of microorganisms (bacteria, archaea, fungi, viruses) that reside in the gastrointestinal tract. In healthy adults, the gut microbiota is dominated by bacteria from the phyla *Firmicutes* and *Bacteroidetes*, with lesser contributions from *Proteobacteria*, *Actinobacteria*, *Verrucomicrobia*, and others [13]. These microbes are not merely commensals, rather they fulfill numerous essential functions critical for host homeostasis. Collectively, the gut microbiota harbors an extensive genetic repertoire that endows humans with metabolic capabilities beyond those encoded by the human genome [13]. For instance, dietary components that are indigestible by human enzymes (like certain fibers and polyphenols) can be broken down by gut bacteria into absorbable and bioactive molecules. In this way, the gut microbiome acts as an “organ” that extends our normal physiology, often termed the “second genome”, or an endocrine organ of the host [14].

A key concept in microbiome science is that gut microbes produce a wide range of metabolites, which enter the circulation and influence distant organs. These metabolites can be classified by their origin [15]: (1) diet-derived microbial metabolites, which are compounds generated when gut bacteria ferment or transform dietary substrates. A prime example is short-chain fatty acids (SCFAs), such as acetate, propionate, and butyrate, produced from fiber fermentation. SCFAs serve as energy sources for colonic cells and also act as signaling molecules that modulate inflammation and metabolism throughout the body. Butyrate, in particular, has anti-inflammatory effects and supports the integrity of the gut barrier. Other diet-derived metabolites include certain polyunsaturated fatty acid byproducts and amino acid catabolites (e.g., indole derivatives from tryptophan) [16]. (2) De novo microbial metabolites: These are molecules synthesized by microbes that are not directly derived from the diet. Lipopolysaccharide (LPS) is a notable example; it is a component of the outer membrane of Gram-negative bacteria and can enter circulation if the intestinal barrier is breached. LPS is a potent endotoxin that triggers systemic inflammation. Another example is trimethylamine (TMA), which gut bacteria produce from dietary choline or carnitine; the host liver then converts TMA into trimethylamine-N-oxide (TMAO). TMAO has garnered attention for its association with cardiovascular disease risk and oxidative stress. Additionally, gut microbes synthesize essential nutrients like vitamin K and certain B vitamins de novo [17]. (3) Host–microbial cometabolites: These originate from host compounds that are modified by the microbiota. A prominent category is secondary bile acids; after the liver secretes bile acids into the gut to aid digestion, gut bacteria can chemically modify them, producing secondary bile acids that have distinct signaling functions (some of which can activate or inhibit host receptors, affecting metabolism and immune function). Similarly, microbes produce enzymes like hydroxysteroid dehydrogenases that can modify steroid hormones in the gut, affecting their reabsorption or excretion [18]. For example, the estrobolome refers to gut microbial genes capable of deconjugating estrogens, thereby influencing circulating estrogen levels. In men, such microbial transformations could potentially alter the androgen/estrogen balance via the enterohepatic recycling of hormones [19].

An emerging conceptual framework, the “androbactome”, has been proposed to describe the subset of gut microbial genes and species involved in regulating androgen metabolism [12]. Analogous to the well-studied estrobolome, the androbactome encompasses microbes capable of modulating testosterone and other androgens through enzymatic activities, such as β-glucuronidase-mediated deconjugation and 20β-hydroxysteroid dehydrogenase activity [20]. These microbial functions may influence systemic androgen availability, may impact the androgen-to-estrogen ratio, and may potentially affect spermatogenesis and sexual function [12]. Although the term remains largely theoretical, it offers a useful lens for exploring the microbiome’s role in male reproductive endocrinology.

Through these metabolites and other mechanisms, the gut microbiota exerts systemic effects on nearly every organ system [15]. It helps regulate immune maturation and immune responses (about 70% of the body’s immune cells reside in the gut-associated lymphoid tissue) [21]. It influences metabolic health by interacting with host signaling pathways that control glucose and lipid metabolism. It even affects the nervous system via the gut–brain axis, where microbial metabolites (like certain neurotransmitter precursors) and vagus nerve signaling can alter brain chemistry and behavior [22]. Given the profound systemic effects of the gut microbiota, including its influence on immunity, metabolism, and hormone regulation, it is plausible that microbial imbalances may also perturb the male reproductive axis. The following section explores specific mechanistic pathways through which gut dysbiosis may impair testicular function and fertility.

## 4. Mechanistic Insights into the Gut Microbiota–Testis Axis

### 4.1. Immune and Inflammatory Pathways

Dysbiosis of the gut microbiota can induce systemic inflammation that extends to the male reproductive tract. A central mediator is bacterial endotoxin (lipopolysaccharide, LPS), a component of the cell wall of Gram-negative bacteria [23,24]. When dysbiosis compromises the intestinal barrier, LPS translocates into the circulation and binds to Toll-like receptor 4 (TLR4) on immune cells, triggering the release of pro-inflammatory cytokines (e.g., TNF-α, IL-6, IL-1β) that disrupt reproductive tissues [25,26]. LPS from *Escherichia coli* has been shown to stimulate IL-17A production in healthy male mice, contributing to autoimmune-like inflammation in the testes, epithelial cell injury, and reduced sperm output and quality [27,28,29].

In the testes and epididymis, elevated cytokine levels (IL-6, TNF-α, IL-1β) directly reduce sperm motility and viability, while promoting apoptosis, DNA fragmentation [30,31], and the downregulation of β-defensins, which normally enhance sperm integrity [32,33]. For example, LPS-induced epididymitis downregulates sperm-associated antigen 11E (SPAG11E), a rat-specific β-defensin, destabilizing the blood–epididymal barrier and worsening sperm injury [32,33]. Persistent inflammation can lead to orchitis or epididymitis, directly damaging the seminiferous and epididymal epithelia, reducing sperm counts, and delaying sperm transit [34].

Cytokine-mediated disruption of tight junction proteins in the blood–testis and blood–epididymis barriers facilitates immune cell extravasation into these compartments, undermining their protective immunological privilege [35,36]. Prolonged barrier compromise exposes germ cells to autoimmune attack, exacerbating tissue damage [36,37]. Animal models further suggest subacute LPS exposure disrupts testicular architecture, peaking by the end of the spermatogenic cycle; conversely, vitamin K supplementation can dampen inflammatory pathways and support testosterone biosynthesis [38,39]. These findings underscore gut–testis immune crosstalk, wherein gut-derived triggers can impair hormonal output and germ cell health [40]. A complementary model by Zeng et al. (2025) demonstrated that gut-derived Th17 cells, activated by intestinal inflammation, can traffic to the testes and cause autoimmune orchitis, providing further evidence that immune dysregulation along the gut–testis axis may impair spermatogenesis [41].

### 4.2. Oxidative Stress and Antioxidant Deficiency

An imbalance in gut flora can elevate systemic oxidative stress (OS) levels [42,43]. Endotoxin-driven inflammation recruits neutrophils and macrophages to the male reproductive tract, generating excessive reactive oxygen species (ROS) [44,45]. Because sperm cells contain high concentrations of polyunsaturated fatty acids but have limited antioxidant capacity, they are highly vulnerable to ROS-mediated lipid peroxidation, protein, and DNA damage, leading to reduced motility, viability, and fertilization capacity [46,47]. Moreover, mitochondrial damage from LPS-induced oxidative stress further impairs sperm function [48].

In support of these findings, clinical evidence also suggests that specific natural compounds can directly ameliorate mitochondrial oxidative damage in sperm. A pilot study by Illiano et al. demonstrated that a resveratrol-based multivitamin supplement significantly increased the sperm concentration and motility in men with idiopathic infertility, potentially through antioxidant actions targeting mitochondrial function and ROS reduction [49]. This adds to the growing literature supporting the use of targeted nutraceuticals in modulating the redox balance at the level of spermatozoa.

Certain gut microbes synthesize or support the absorption of antioxidants (e.g., selenium, glutathione). Probiotic strains can increase host antioxidant enzyme levels, such as SOD, GPX, and PRDX, to mitigate ROS toxicity [50,51,52]. For instance, peroxiredoxin 6 (PRDX6) helps maintain sperm DNA integrity via the phosphoinositide 3-kinase/protein kinase B (PI3K/AKT) pathway [51,52], while nuclear factor erythroid 2-related factor 2 (Nrf2)-activating Lactobacillus species can upregulate endogenous antioxidant genes [heme oxygenase-1 (HO-1), catalase, SOD] [53]. Supplementation with antioxidants [54,55] or probiotics [56,57] improves sperm quality by reducing oxidative damage and reinforcing sperm DNA integrity. Selenium supplementation, in particular, boosts antioxidant enzymes and beneficially reshapes the gut microbial composition, increasing Lactobacillus and Christensenella, while reducing pro-inflammatory taxa (Rikenella, Alistipes), ultimately improving sperm parameters [58,59].

Seminal oxidative stress, often detected due to elevated malondialdehyde (MDA) and 8-hydroxy-2′-deoxyguanosine (8-OHdG), is common in infertile men [60]. In obese infertile men, circulating LPS-binding protein (LBP), a marker of endotoxemia, positively correlates with 8-OHdG in sperm DNA, suggesting a gut–ROS–sperm damage axis [60]. Dysbiosis-induced IL-17A expression further activates oxidative damage in testes, lowering sperm vitality and raising deformity rates [61]. Meanwhile, dysbiosis may diminish nutrient absorption (e.g., zinc, vitamin C, folate) [62] and reduce co-factors essential for glutathione synthesis and DNA repair (e.g., vitamin B12, selenium) [63]. Elevated trimethylamine N-oxide (TMAO), linked to high meat diets and dysbiotic microbiota, may also exacerbate testicular oxidative injury, although direct evidence in regard to male fertility remains limited [64].

### 4.3. Metabolites of Gut Microbiota

Gut microbial metabolites, such as short-chain fatty acids (SCFAs), secondary bile acids, tryptophan/indole derivatives, and vitamins, are key effectors linking microbiota composition to sperm health [34]. SCFAs (acetate, propionate, butyrate) modulate lipid and glucose metabolism, reduce inflammation, and enhance spermatogenesis [65,66]. Notably, butyrate improves sperm count, motility, and intratesticular testosterone in animal models, while also boosting seminal GPx and SOD activity [67,68]. It can act as a histone deacetylase (HDAC) inhibitor, promoting the transcription of antioxidant and metabolic genes that are vital for spermatogenesis and strengthening the blood–testis barrier via tight junction upregulation [69].

Secondary bile acids, formed by the microbial transformation of primary bile acids, influence lipid metabolism and immune responses [70]. The dysregulation of bile acids can impair steroid and retinoid metabolism, leading to testicular injury and defective spermatogenesis [71]. Specific fungal components, such as Aspergillus fumigatus, may enhance bile salt hydrolase-expressing bacteria, thereby increasing retinol absorption and improving testicular retinoid levels [72]. Similarly, tryptophan-derived indoles (e.g., indole-3-propionic acid) protect the testis through antioxidant and anti-inflammatory activities, supporting testosterone synthesis and mitigating oxidative damage [73,74,75,76]. Indoles also act via the aryl hydrocarbon receptor (AHR) in Sertoli and Leydig cells, modulating inflammation and barrier function [76].

Certain gut bacteria (e.g., *Bifidobacterium*, *Lactobacillus*) synthesize vitamin K or produce folate, contributing to DNA methylation in rapidly dividing spermatogonia [77,78]. Paternal folate deficiency can lead to imprinting defects, emphasizing the role of gut flora in ensuring adequate folate bioavailability [79,80]. An “androbactome” concept has been proposed, parallel to the female “estrobolome,” describing microbial taxa and metabolites that modulate androgen metabolism via enterohepatic recycling [53]. Dysbiosis may thus alter androgen-to-estrogen ratios, indirectly affecting spermatogenesis. In regard to a seasonal model, Wu et al. (2025) showed that testicular polyamine levels fluctuate in response to gut microbiota shifts, modulating spermatogenesis through microbial–metabolite crosstalk [81].

### 4.4. Epigenetic Modifications

Gut microbiota shape the epigenetic landscape of male germ cells by modulating DNA methylation, histone modifications, and non-coding RNA expression [82,83]. One route involves the microbial production of folate and other B vitamins, which are essential for one-carbon metabolism and S-adenosylmethionine (SAM) synthesis; depleted folate or vitamin B12 levels can cause aberrant DNA methylation in sperm [84,85]. Dysbiosis may also lower SAM levels via impaired choline metabolism [86].

Short-chain fatty acids (SCFAs), such as butyrate, can inhibit histone deacetylases (HDACs) [87], altering the chromatin structure and gene expression during spermatogenesis [88]. Microbial signals further influence the testicular microRNAs involved in meiosis, chromatin remodeling, and small RNAs packaged into sperm, potentially contributing to transgenerational inheritance [89]. GM-targeted interventions (e.g., black tea extract supplementation) have been shown to shift the gut microbiota composition and change the methylation of imprinted genes in sperm [15]. These epigenetic alterations may compromise fertility and even affect offspring health by disrupting embryonic gene expression [90]. Thus, the gut “microbiome–epigenome” axis links lifestyle factors to male reproductive potential.

### 4.5. Hormonal Regulation via the HPG Axis

Gut microbes profoundly affect the hypothalamic–pituitary–gonadal (HPG) axis, influencing the synthesis of testosterone, LH, and FSH [91]. Chronic endotoxemia due to dysbiosis can suppress GnRH release and reduce LH/FSH pulsatility, causing lower Leydig cell testosterone output and impaired spermatogenesis [92]. This phenomenon, encapsulated by the “Gut Endotoxin Leading to Decline in Gonadal Function” (GELDING) theory, proposes that LPS from a leaky gut drives systemic inflammation, blunting hypothalamic–pituitary signaling [93].

SCFAs, such as butyrate, can stimulate GLP-1, improving insulin sensitivity and boosting testosterone levels [94,95]. GLP-1 analogs used in treating type 2 diabetes have also been observed to increase serum testosterone and improve semen parameters [96]. Certain microbial strains (e.g., *Clostridium scindens*) convert glucocorticoids into androgenic precursors, while β-glucuronidase-expressing microbes facilitate enterohepatic recycling of sex steroids [97,98]. Dysbiosis can favor estrogenic metabolites, lowering bioavailable testosterone and harming male fertility [99]. Men with more diverse gut microbiota, including Bacteroides, Dorea, and Ruminococcus, tend to have higher testosterone [100], whereas low testosterone is associated with opportunistic pathogens and Firmicutes depletion [101]. Notably, sex steroids also shape the gut microbial composition, indicating a bidirectional gut–testis feedback loop [102]. A 2024 study identified a testosterone-degrading bacterium (*Pseudomonas nitroreducens*), whose colonization in mice reduced the level of circulating testosterone, highlighting the potential of specific microbial enzymes in regard to modulating systemic androgen availability [103].

### 4.6. Blood–Testis Barrier Modulation

The blood–testis barrier (BTB) provides an immunoprivileged environment for developing germ cells. However, dysbiosis-related inflammation elevates cytokine levels, such as IL-6 and TNF-α, which disrupt Sertoli cell tight junctions and increase BTB permeability [104,105]. IL-6 alters the expression of key tight junction proteins (occludin, claudin-11, N-cadherin), weakening their structural integrity and allowing immune cell infiltration [106]. Autoimmune reactions may ensue, harming sperm production [107].

Butyrate and other SCFAs support epithelial barrier function by upregulating tight junction protein expression, and experimental models show that probiotics or SCFA supplementation can preserve BTB integrity under inflammatory and metabolic stress [108,109]. Adequate testosterone levels, partly modulated by gut microbiota, also stabilize the BTB and aid its normal restructuring during spermatocyte transit [109,110]. Chronic BTB disruption can result in sperm granulomas and long-term reductions in sperm output, emphasizing the importance of gut homeostasis for testicular immune privilege [36]. These findings underscore the promise of microbiota-centered strategies in tackling the global decline in sperm quality and enhancing male reproductive outcomes (Figure 2).

## 5. Therapeutic and Lifestyle Interventions

### 5.1. Probiotics and Prebiotics

One of the most direct ways to favorably alter the gut microbiota is through probiotics (beneficial live microorganisms administered as a supplement) or prebiotics (indigestible fibers that nourish beneficial gut bacteria). Over the past decade, several studies have examined whether probiotics or prebiotic supplementation can improve semen parameters in animals and humans. The results have been promising, indicating that these interventions can mitigate dysbiosis-related damage to the male reproductive system [111,112].

In animal models, numerous trials demonstrate reproductive benefits of probiotics. For example, giving *Lactobacillus rhamnosus* to diabetic rats led to higher testosterone levels and increased sperm counts compared to untreated diabetic rats, likely by reducing gut inflammation and oxidative stress [113]. Similarly, supplementation with a cocktail of *Lactobacillus* and *Bifidobacterium* strains in obese mice improved sperm motility and reduced testicular inflammatory cytokines [114]. Prebiotics, which include fibers like inulin, fructooligosaccharides (FOS), resistant starch, and specialized oligosaccharides, have also shown efficacy. Alginate oligosaccharide (AOS), a prebiotic derived from algae, was used in a mouse study of high-fat diet-induced infertility: AOS supplementation reshaped the gut microbiome, increasing SCFA producers, and as a result improved the mice’s sperm concentration, viability, and testis morphology [115]. AOS-treated mice had less gut permeability and endotoxin load, linking the prebiotic’s effect to the mechanistic pathways we described. These animal studies firmly establish that targeted microbiome nourishment can translate into tangible improvements in male reproductive metrics. Similarly, Ferenczi et al. (2025) showed that *Lactiplantibacillus plantarum* supplementation enhanced the semen volume, sperm concentration, motility, and testicular antioxidant status in a poultry model, reinforcing the probiotic potential across species [116].

Recent clinical trials have mirrored these findings in infertile men. A systematic review in 2024 compiled data from multiple randomized controlled trials (RCTs) on probiotic use in regard to male infertility [117]. The consensus was that probiotics can significantly improve sperm parameters in men with idiopathic infertility. For instance, Maretti & Cavallini (2017) conducted a double-blind trial in which one group of infertile men received an oral probiotic blend (*Lactobacillus*, *Bifidobacterium*, and *Streptococcus thermophilus*) for 6 months. Men in the intervention group exhibited significant improvements in their sperm concentration (from ~20 million/mL to ~40 million/mL on average) and motility, along with a decrease in abnormal forms, compared to the placebo [118]. Another RCT by Helli et al. (2022), involving 50 infertile men, used a 7-strain probiotic formula for 10 weeks and reported a significant rise in sperm count (+≈15 million/mL) and progressive motility in the probiotic group, whereas the placebo group showed no improvement [119]. Importantly, Helli et al. also measured oxidative stress markers and found that the probiotic-treated men had substantially lower semen MDA levels and a higher total antioxidant capacity, indicating that the probiotic restored the redox balance in the semen [119]. This aligns with mechanistic expectations, namely that by reducing gut-sourced oxidative stress, the sperm quality improved.

Probiotics may also positively impact hormonal profiles in infertile men. In the aforementioned trials, some noted modest increases in serum testosterone and gonadotropins after probiotic therapy [117]. For example, Maretti and Cavallini observed a rise in testosterone and a reduction in estradiol in the probiotic group, shifting the hormone balance in favor of spermatogenesis [118]. While these hormone changes were not large, they point to the endocrine modulation possible via microbiome alteration.

Prebiotic interventions in humans are less frequently reported than probiotics, but conceptually they achieve a similar end: nurturing beneficial bacteria. Some studies have used synbiotics (probiotic + prebiotic combination), making it hard to isolate the prebiotic contribution. However, diets naturally high in prebiotic fibers (fruits, vegetables, whole grains) correlate with better semen quality in observational studies [120]. We can extrapolate that increasing dietary fiber or taking a prebiotic supplement fosters SCFA-producing gut flora, which in turn might reduce systemic inflammation and improve metabolic parameters relevant to fertility. One trial in subfertile men tested a synbiotic (a product with multiple probiotic strains plus fructooligosaccharide) for 3 months and found improved sperm DNA integrity and chromatin quality in the treated group [121]. The fiber likely helped sustain the introduced probiotic strains and enhance butyrate production.

When considering specific strains, those from the genera *Lactobacillus* and *Bifidobacterium* are the most commonly used and appear effective [117]. These are lactic acid-producing bacteria that can lower the intestinal pH and outcompete pathogenic bacteria. They also have known anti-inflammatory effects on the gut lining. Strains like *L. rhamnosus*, *L. casei*, *L. acidophilus*, *B. longum*, and *B. breve* have all been included in successful trials [117]. Notably, these strains are generally recognized as safe and are already present in fermented foods and a healthy human gut. The doses used in studies range from ~1 billion to 10+ billion colony forming units (CFUs), daily.

For clinicians and patients, probiotic therapy for male infertility is attractive because it is relatively inexpensive, safe, and non-invasive compared to assisted reproductive techniques. However, it is important to set realistic expectations: probiotics are not a panacea and likely benefit a subset of patients (particularly those with evidence of oxidative stress or inflammation) [122]. They may be most useful as an adjunct to other treatments or as primary therapy in men with mild idiopathic infertility.

Prebiotics can be incorporated easily through the diet, e.g., increasing dietary fiber (vegetables, legumes, whole grains) and resistant starch (cooked then cooled potatoes or rice), or through supplements like psyllium husk, inulin, etc. These changes improve gut microbiome diversity and SCFA output. As a side benefit, a high-fiber diet also aids weight management and insulin sensitivity, which themselves improve the hormonal milieu that is beneficial for fertility [123].

### 5.2. Dietary Modifications

Diet is one of the most powerful levers to shape the gut microbiome and, by extension, systemic health. As discussed earlier, dietary patterns strongly correlate with semen quality and much of that influence likely occurs through microbiome-mediated pathways. Therefore, dietary modification is a cornerstone intervention for improving male fertility in the context of the gut–testis axis [124,125].

A whole-food, plant-rich diet has consistently been associated with better sperm parameters. Diets such as the Mediterranean diet, which emphasize fruits, vegetables, whole grains, legumes, fish, and olive oil (with limited processed foods and red meat), provide abundant fibers and polyphenols that feed beneficial gut bacteria. These diets are anti-inflammatory and antioxidant rich. In a systematic review of 10 studies involving ~2000 men, adherence to the Mediterranean diet significantly correlated with a higher sperm concentration and motility [126]. The likely mechanism is multifaceted: the diet improves overall metabolic health, but also increases gut microbiota diversity and SCFA production due to its high fiber content. SCFAs (like butyrate) from fiber fermentation enforce gut barrier integrity and reduce systemic inflammation, thereby creating a more favorable environment for spermatogenesis. The diet’s high omega-3 fatty acids content (from fish, nuts) also modulates the gut microbiome and has direct benefits on sperm membrane composition and motility. Indeed, men on Mediterranean diets show lower levels of inflammatory markers and oxidative stress, which align with improved sperm function [126,127].

Conversely, the Western diet, characterized by a high intake of red/processed meats, refined carbohydrates, sugary beverages, and saturated or trans fats, is associated with poorer semen quality. Men who consume a Western-style diet tend to have lower sperm motility and morphology percentages, on average [128]. This diet promotes an obesity-prone microbiome, often increasing bile-tolerant, LPS-containing bacterial groups, and decreasing beneficial fiber degraders. High fat and sugar intake can lead to gut dysbiosis, with increased permeability (as noted, dietary fats can emulsify gut membranes facilitating endotoxin absorption). Additionally, such diets often lack micronutrients, like vitamin C, folate, and zinc, that are crucial for spermatogenesis and are also needed by beneficial gut microbes. The end result is a pro-inflammatory, oxidative internal milieu antagonistic to fertility [129]. However, the damage from a Western diet appears reversible; studies show that switching to a high-fiber diet can restore a healthier microbiome within weeks. Weight loss through diet (and exercise) also reduces circulating endotoxin and inflammatory cytokines, which has been linked to improved testosterone levels and sperm counts in obese men [130,131].

Specific dietary components that deserve a mention:

Dietary Fiber: Aside from overall patterns, total fiber intake has been singled out as beneficial. Fiber feeds SCFA-producing bacteria (like *Roseburia*, *Faecalibacterium*), leading to more butyrate and propionate in circulation. These SCFAs can modulate immune responses and even influence sperm metabolism (propionate can be an energy source) [132]. A higher fiber intake is associated with lower C-reactive protein (CRP) and IL-6 in men, indicating reduced inflammation that would benefit testicular function. Clinically, men advised to increase their daily fiber intake (e.g., consuming >30 g/day from diverse sources) showed slight improvements in sperm motility in some trials, although more research is needed [133].

Polyphenols: These are bioactive compounds in colorful fruits, vegetables, tea, coffee, and wine (e.g., flavonoids, catechins, resveratrol). Polyphenols are poorly absorbed in the small intestine and, thus, reach the colon, where gut microbes metabolize them into bioactive metabolites (like urolithins from pomegranate, or equol from soy isoflavones). Polyphenol-rich foods promote the growth of beneficial bacteria, such as *Bifidobacteria* and *Akkermansia* [134]. In turn, these microbes can produce anti-inflammatory compounds. Polyphenols themselves (and their metabolites) can scavenge ROS and improve endothelial function. In regard to male fertility, polyphenol intake (from berries, citrus, green tea, etc.) has been linked to better sperm DNA integrity and higher progressive motility in observational studies. One example is green tea catechins, which were shown to improve the sperm count in rodents, while also altering the gut flora composition favorably [135]. Thus, including a variety of polyphenol-rich plant foods in the diet likely benefits the gut–testis axis.

Fat Quality: Replacing saturated fats with unsaturated fats (especially omega-3 from fish and flaxseed) can reduce gut endotoxin levels. Omega-3 fatty acids have anti-inflammatory effects and can increase certain beneficial gut bacteria. Moreover, omega-3s are incorporated into sperm membranes, enhancing fluidity and viability. Diets deficient in omega-3 or high in trans fats have been tied to lower sperm concentrations. Given that gut microbes can modulate bile acid profiles (which affect fat absorption and cholesterol metabolism), improving the fat quality in the diet also interacts with the microbiome. Some studies gave men fish oil supplements (rich in omega-3 EPA/docosahexaenoic acid [DHA]) and found improved sperm morphology; interestingly, fish oil also tends to increase *Lactobacillus* in the gut [136].

Micronutrients and Fermented Foods: A diet naturally high in antioxidants (vitamins C, E, beta-carotene, selenium, etc.) supports both microbiome health and sperm quality. Many antioxidant-rich foods are also prebiotic (fruits, leafy greens). Including fermented foods, such as yogurt, kefir, kimchi, and sauerkraut, can introduce live probiotics and substrates that benefit gut health. Some studies have seen that men who regularly consume yogurt have higher sperm motility than those who do not, potentially due to its lactic acid bacteria content [137,138].

An emerging area of interest is the role of environmental chemicals in food (like pesticide residues, microplastics) on the gut–testis axis. For example, the chronic ingestion of low-dose pesticides could alter the gut microbiome in a way that exacerbates their endocrine-disrupting effects. One study found that probiotics could ameliorate sperm damage caused by microplastic exposure in mice, hinting that diet and the microbiome can offset some toxic exposures [139]. This is an area for future research, but it underscores the diet’s role not just in providing nutrients, but also in mitigating harms.

### 5.3. Fecal Microbiota Transplant (FMT)

Fecal microbiota transplant, the process of transferring stool (and, thus, the entire microbial community) from a healthy donor to a patient’s gastrointestinal tract, is an advanced intervention that has shown dramatic efficacy in treating certain microbiome-related diseases (like *Clostridioides difficile* infection). In the context of male infertility, FMT is still experimental, but animal studies provide a proof-of-concept for its use in restoring fertility compromised by dysbiosis [140,141].

As highlighted earlier, an FMT from healthy donors into infertile mice (infertile due to a high-fat diet or other insults) was able to rescue sperm production and quality. In one case, mice with chemotherapy-induced damage to spermatogenesis had significantly improved sperm counts after receiving an FMT from robust, fertile mice [142]. A study by Hao et al. (2022) introduced an alginate oligosaccharide-conditioned donor microbiota (A10-FMT), which restored the sperm counts, motility, and metabolic parameters in high-fat diet-induced subfertile mice [143]. The donor microbiome likely contained beneficial taxa and functional genes that the recipient lacked. By recolonizing the recipient’s gut, an FMT replenished the SCFA levels, reduced systemic toxins, and normalized the metabolism (for instance, lowering blood glucose in diabetic infertile mice) [144]. This led to a stronger “systemic environment” for sperm development, namely improved liver and spleen function (suggesting reduced inflammation) and increased levels of key testicular metabolites like DHA and testosterone [145,146]. In effect, an FMT reset multiple axes, metabolic, immune, and endocrine, back to a state supportive of fertility.

While most current evidence stems from animal models, preliminary insights may inform future clinical strategies. Translating FMT to human male infertility is challenging and, to date, largely hypothetical. FMT, while promising in preclinical models, is not yet approved for use in regard to male infertility and remains a research tool rather than a standard therapy. Its future clinical application would require rigorous evaluation of long-term safety, donor selection protocols, and ethical considerations, before it can be implemented outside controlled trials. There are ethical and practical considerations: FMT involves introducing another person’s microbiome into the patient, with risks of transferring infections or unknown microbes. It is usually reserved for life-threatening or severe conditions (e.g., refractory C. diff colitis) [147]. However, one could envision trials in the future for specific cases of male infertility, particularly if linked to a metabolic syndrome or severe dysbiosis, when conventional therapies have failed. For example, an obese man with hypogonadism and low sperm count might benefit from FMT from a lean, healthy donor as a way to rapidly shift his microbiome and metabolic profile [148,149]. Some indirect evidence comes from bariatric surgery literature; men who undergo a gastric bypass (which alters the gut microbiome substantially) often have improved testosterone and sperm counts post-surgery, hinting at a microbiome link [150].

If FMT were to be tried in regard to male infertility, strict screening of donors would be paramount to avoid transmitting pathogens. The donor would ideally be someone with excellent semen quality and metabolic health, hypothesizing that their microbiome harbors “fertility-promoting” microbes [151]. One could also consider an autologous FMT approach: for instance, a man’s own microbiome was banked when he was younger or healthier and re-infused later if fertility issues arose [152]. This is speculative, but as FMT science advances, capsulized and refined microbiota preparations (sometimes called “microbiota therapy”) might become safer and more targeted.

Another avenue is using FMT in research to identify which microbes are key. If FMT from a particular donor consistently improves fertility in animal models, researchers can analyze that donor’s stool to pinpoint the candidate organisms or metabolites responsible. Those could then be developed into probiotic formulations or drugs, obviating the need for full FMT [153]. In essence, FMT serves as a discovery tool for fertility-enhancing microbes.

It is worth noting that FMT could have unintended hormonal effects; since the gut contributes to hormone regulation, a new microbiome might acutely alter estrogen or cortisol metabolism. While this might be beneficial (as in Tremellen’s theory, reducing endotoxin should raise testosterone), careful monitoring would be required [154].

### 5.4. Other Lifestyle Factors

Beyond diet and direct microbiome therapies, several other lifestyle factors can influence the gut microbiota and, thereby, male fertility. Attention to these factors provides a holistic approach to optimizing the gut–testis axis.

Physical Activity: Regular exercise is beneficial for sperm production and hormonal health [155]. One mechanism that helps to achieve this benefit is through the microbiome. Studies have shown that consistent aerobic exercise increases gut microbial diversity and boosts populations of butyrate-producing bacteria, while reducing pro-inflammatory bacteria. Exercising rodents have a more robust gut barrier and lower endotoxin levels than sedentary ones [156,157]. In men, moderate exercise has been associated with higher antioxidant capacity in semen and improved sperm motility, possibly due to reduced systemic inflammation. Exercise also helps maintain a healthy body weight, thereby reducing obesity-related dysbiosis [158]. However, it is important to avoid excessive endurance training without adequate recovery, as overtraining can transiently raise cortisol levels and gut permeability (extreme endurance athletes sometimes experience “leaky gut” and inflammation) [159]. The key is balanced, regular exercise (e.g., 30–60 min of moderate activity most days), which likely cultivates a microbiome that is metabolically fit and anti-inflammatory [160].

Psychological Stress Management: Chronic stress can negatively impact male fertility due to hormonal suppression (via elevated cortisol and adrenaline) and immune dysregulation. A less appreciated consequence of stress is its effect on the gut. Stress hormones can alter gut motility, mucus secretion, and even modulate the microbiota composition [161]. Cortisol release under stress has been shown to increase gut permeability and disrupt the gut barrier [161], thereby potentially increasing endotoxin passage. Stress can also reduce beneficial Lactobacilli and increase opportunistic pathogens in the gut. These changes together create a pro-inflammatory state [162]. Indeed, animal studies have found that subjecting male rats to chronic psychological stress results in both gut barrier defects and reduced sperm counts and testosterone [163,164]. Techniques to manage stress, such as mindfulness, adequate sleep, counseling, and avoiding chronic occupational stress, may thus indirectly protect fertility by preserving gut health. Ensuring sufficient sleep is also crucial, as circadian rhythm disruptions can perturb the microbiome and hormone release [165]. In essence, caring for one’s mental health and stress levels should be part of fertility care, with the gut–brain axis providing one route by which stress relief translates into reproductive benefits.

Judicious Antibiotic Use: Antibiotics can cause profound short-term disturbances in the gut microbiome. Broad-spectrum antibiotics in particular may wipe out beneficial strains and promote overgrowth of yeasts or resistant bacteria [166,167]. There have been case reports of temporary infertility in men following long courses of antibiotics, which might be partly due to microbiome disruption (and partly direct drug toxicity to sperm in some cases) [168]. If an antibiotic is medically necessary, men trying to conceive should ensure they repopulate their gut with probiotics or fermented foods afterward. Unnecessary use of antibiotics (e.g., for viral infections) should be avoided, not only to prevent resistance, but to preserve microbiome integrity [169]. Notably, certain antibiotics (like tetracyclines or nitrofurantoin) can directly reduce semen quality, but even those without direct effects might indirectly affect fertility via dysbiosis [170]. Therefore, antibiotic stewardship is surprisingly relevant in the context of fertility.

Avoiding Microbiome Disruptors: Aside from antibiotics, other substances can alter the gut flora. Frequent use of NSAID painkillers can increase gut permeability [171]. Excessive alcohol intake is known to cause dysbiosis and a leaky gut, contributing to the concept of “alcoholic endotoxemia”, which can lower testosterone and shrink the testes in extreme cases. Thus, limiting alcohol to moderate levels (if at all) can help maintain a healthier gut microbiome [172]. Smoking is another factor; smoking has been linked to altered gut flora and is a known cause of oxidative stress in semen. The cessation of smoking can improve both the microbiome and sperm health [173].

Environmental Hygiene and Partner’s Microbiome: Interestingly, sexual partners share microbes; partners tend to have more similar microbiomes than random individuals, likely due to intimate contact and a shared diet/environment [11]. If a female partner has suboptimal vaginal or gut microbiota, it could in theory affect the male’s microbiome over time. Some researchers have suggested that treating both partners’ microbiomes might yield the best outcome for a couple’s fertility [174]. For example, if a female has bacterial vaginosis (a microbiome imbalance in the reproductive tract) and the male has dysbiosis, a coordinated probiotic regimen could benefit both, reducing reinfection or cross-seeding of unwanted bacteria. This is a frontier area, but underscores that we should consider the couple as a unit when addressing the microbiome and fertility.

Taken together, the therapeutic interventions explored in this section, ranging from probiotics and dietary strategies to fecal microbiota transplantation and lifestyle optimization, highlight the multifactorial potential of modulating the gut microbiota to improve male reproductive outcomes. To consolidate these findings, Table 1 presents an integrated overview of the proposed interventions, their mechanistic basis in regard to the gut microbiota–testis axis, and their documented effects on sperm parameters and hormonal balance.

## 6. Future Directions and Research Gaps

The recognition of a gut microbiota–testis axis in male infertility is relatively recent, and, accordingly, many questions remain unanswered. As research progresses, several key future directions and gaps need to be addressed:

Longitudinal and Causal Studies: Most human data on the matter, so far, are cross-sectional. There is a need for longitudinal studies tracking young men’s gut microbiome over time to see whether changes predict later declines in sperm quality. Such studies could establish temporality and causality, for example, does a shift toward a pro-inflammatory microbiome precede the development of oligospermia? Animal models can only go so far; ultimately, longitudinal human cohorts or interventional trials are needed to confirm that altering the microbiome leads to improved fertility outcomes (natural conception rates, not just semen parameters). Mendelian randomization analyses have been attempted (using genetic proxies for microbiome traits to infer causality) and some suggest a causal influence of certain gut bacteria on infertility risk [175], but these are preliminary. Controlled trials (e.g., giving a prebiotic versus a placebo for a year to men at risk of infertility) could robustly demonstrate causation.

Microbiome-Based Diagnostics: A Mendelian randomization study by Fu et al. (2023) identified causal associations between gut microbial taxa and male infertility risk, suggesting that certain genera may serve as diagnostic markers for dysbiosis-associated infertility [176]. One exciting avenue is the development of microbiota-based diagnostic tools or biomarkers for male infertility. In the future, a simple stool test might help identify men whose infertility is related to dysbiosis. For instance, a “high endotoxin-producing microbiome” signature or low diversity could flag individuals who would benefit from microbiome-targeted therapies. Specific microbial species (such as an overabundance of *Prevotella* or *Oscillibacter*) might serve as biomarkers for inflammation-related infertility [177]. Additionally, metabolite profiles in blood (like high TMAO or low butyrate levels) could be proxies for gut microbial function and correlate with fertility status [178]. Integrating such tests into routine infertility evaluations could provide a more personalized approach, distinguishing those who might respond to diet/probiotic interventions from those who have other dominant factors. At present, microbiome-based diagnostics and targeted interventions remain investigational, and their incorporation into routine fertility assessment awaits robust validation through prospective clinical trials.

Therapeutic Personalization: If certain bacteria are found to be particularly beneficial (or harmful) for male fertility, this could lead to next-generation probiotics or phage therapies. For example, if *Faecalibacterium prausnitzii* is confirmed to support spermatogenesis via anti-inflammatory effects, one could develop a live biotherapeutic containing this anaerobe to administer to patients [179]. Conversely, if *Prevotella copri* is confirmed as a culprit, a targeted bacteriophage or small molecule might be used to selectively reduce its population in the gut [180]. Essentially, the goal would be to craft microbiome therapies tailored to an individual’s microbial composition, a facet of precision medicine. Such approaches will require extensive microbial genomic analysis and safety testing.

Female and Maternal Factors: While our focus is male infertility, future research might also consider the paternal microbiome’s effect on offspring and female partners. The study by Argaw-Denboba et al. (2024), which showed that paternal microbiome perturbation affected the offspring via small RNAs in sperm, opens up a novel dimension [89]. It suggests that a father’s microbiome might influence the health of the next generation, independent of genetics, essentially a form of non-genetic inheritance. Further research in this area could underscore the importance of men optimizing their microbiome even before conception to ensure healthier children (e.g., avoiding the transmission of negative epigenetic signals) [89]. Additionally, since infertility often involves a couple, exploring how the male and female microbiomes interact (the “reproductive microbiome synergy”) could be valuable [181]. These intersections of male and female microbiome research are largely unexplored.

Addressing Controversies: There are also controversies and confounders to address. Not all studies agree on the extent of sperm decline globally, and not all cases of male infertility will involve microbiome issues. Large population studies could clarify the proportion of idiopathic infertility that might be attributable to inflammation associated with dysbiosis versus other factors (like subtle genetic polymorphisms).

An increasing body of evidence demonstrates that the seminal microbiome, comprising bacteria residing within ejaculate, can influence sperm quality, morphology, and motility. For instance, specific taxa, such as *Prevotella, Gardnerella*, and *Streptococcus,* have been linked to impaired sperm function, inflammation, and elevated oxidative stress [182,183]. These microbial profiles may affect DNA integrity, acrosome function, and motility, potentially via cytokine-mediated mechanisms or ROS production. As such, the semen microbiota should be considered in both diagnostic and therapeutic strategies for male infertility, particularly in idiopathic cases. Future research should aim to establish causal links and determine whether microbiome modulation can restore sperm health.

## 7. Conclusions

The gut microbiota–testis axis highlights the capacity of intestinal microbes and their metabolites to influence male reproductive potential. Current evidence, largely preclinical or derived from small, heterogeneous clinical cohorts, demonstrates associations that remain preliminary and not yet causally established between dysbiosis and impaired spermatogenesis. Chronic inflammation triggered by endotoxin (LPS) leakage, oxidative stress mediated by reactive oxygen species (ROS), disruptions to the hypothalamic–pituitary–gonadal hormonal axis, and compromised intestinal barrier function, all contribute to impaired fertility. Conversely, therapies that restore or maintain a healthy microbiome, such as dietary fiber-rich regimens, probiotics, prebiotics, and possibly fecal microbiota transplantation, have shown promise in mitigating these mechanisms, improving sperm parameters, and enhancing overall reproductive outcomes. Moreover, lifestyle factors like balanced exercise, stress management, antibiotic stewardship, and reduced alcohol consumption can protect the gut from dysbiosis and, thereby, support optimal testicular function.

Future research into the gut microbiota–testis axis should focus on strengthening causal evidence involving humans through longitudinal studies and well-controlled clinical trials, developing microbiome-based diagnostic tools, and exploring personalized therapeutic strategies. Identifying specific bacterial strains or microbial metabolites that promote fertility opens new avenues for next-generation probiotics and precision medicine interventions. Investigations into partner-shared microbiomes and intergenerational effects via epigenetic mechanisms may further expand our understanding of how the gut microbiota influences fertility across entire populations. Ultimately, a deeper appreciation of the gut’s systemic influence offers a vital opportunity to address the global decline in sperm quality by refining both preventive and therapeutic approaches to male infertility.

## Figures and Tables

**Figure 1 ijms-26-06211-f001:**
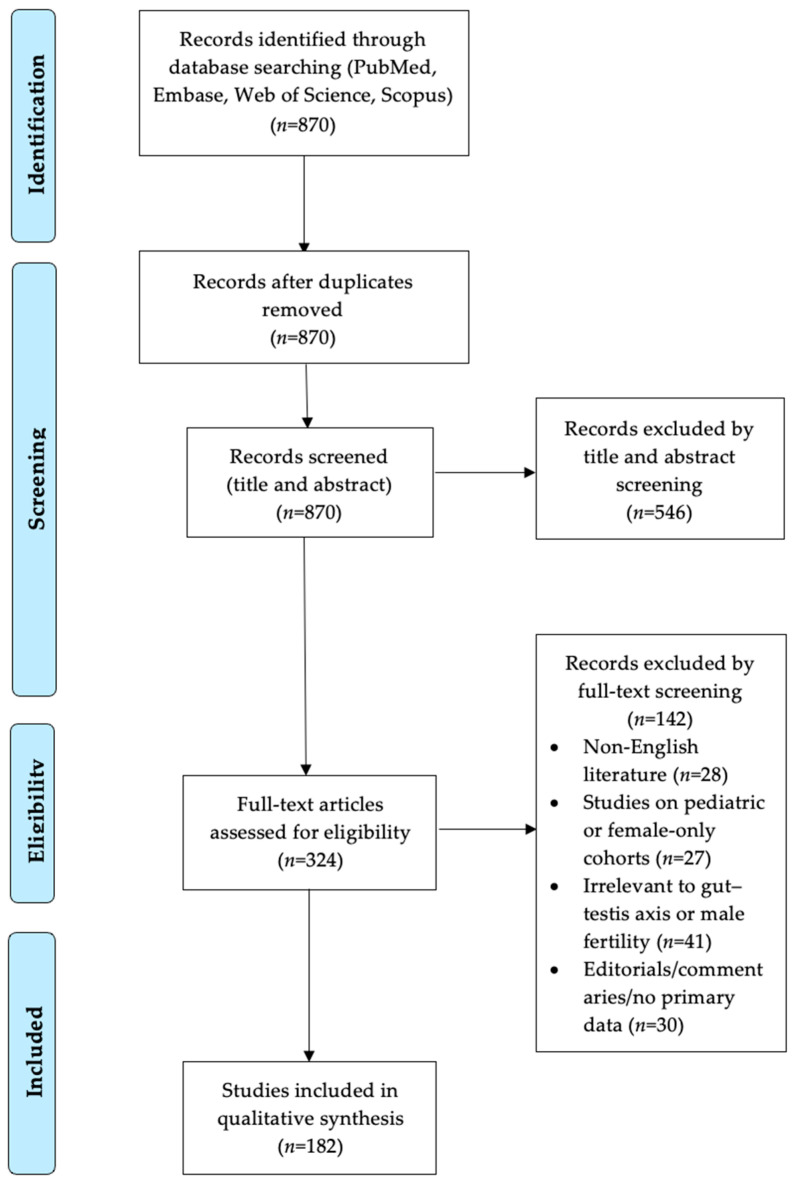
PRISMA-style flow diagram illustrating literature identification, screening, and inclusion steps for the narrative review.

**Figure 2 ijms-26-06211-f002:**
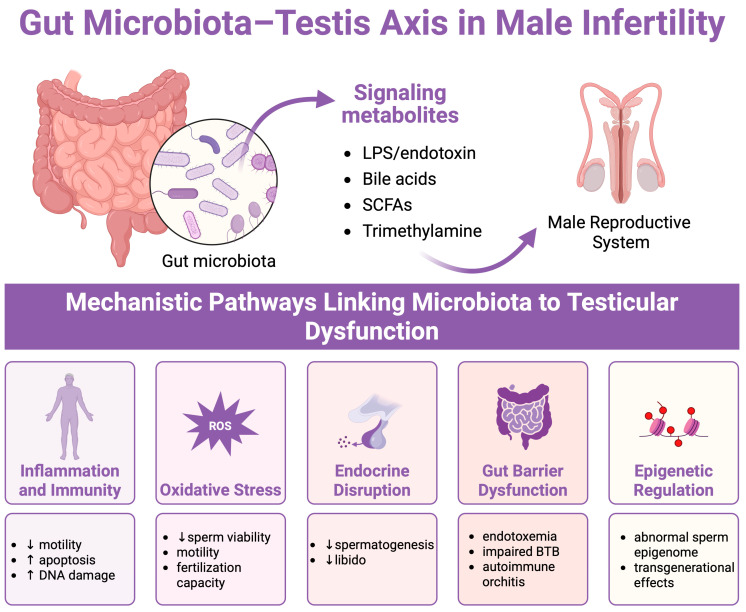
Conceptual overview of the gut microbiota–testis axis in regard to male infertility. Created in BioRender. Kaltsas, A. (2025) https://BioRender.com/s1tl18j.

**Table 1 ijms-26-06211-t001:** Microbiota-targeted and lifestyle interventions shown to improve sperm parameters through the gut–testis axis.

Intervention	Mechanistic Rationale (Gut–Testis Axis)	Reported Effects on Sperm/Male Fertility	Principal Evidence (Discussed in This Narrative Review)
Probiotics	Reduce gut inflammation and oxidative stress Raise host antioxidant enzymes (SOD, GPx, PRDX6) Modulates HPG axis hormones	↑ Sperm concentration and progressive motility ↓ Abnormal morphology ↓ Seminal MDA ↑ Total antioxidant capacity Modest ↑ serum testosterone	Rodent models [113,114] Idiopathic-infertile men, double-blind RCTs [118,119]
Prebiotics	Enriches SCFA-producing taxa Strengthen gut barrier Lower endotoxemia	Mice: ↑ sperm concentration, viability, improved testis histology; ↓ gut permeability Men (synbiotic): improved sperm DNA integrity	Animal studies [123] Triple-blind synbiotic RCT [121]
Mediterranean-Style Diet	High fiber/polyphenols → richer microbial diversity and SCFA output Anti-inflammatory nutrient profile	↑ Sperm concentration, motility, and morphology ↓ Systemic oxidative stress	Systematic reviews and cohort/RCT data [126,127,130,131]
Fecal Microbiota Transplant (FMT)	Restores whole microbial community in severe dysbiosis	Murine models: ↑ spermatogenesis, ↑ intratesticular testosterone, normalized metabolic milieu	Preclinical and translational studies [140,142,144,147,148,149,151,152,153,154]
Exercise and Stress-Reduction Practices	Moderate exercise expands butyrate-producing genera and gut diversity Stress control curbs cortisol-driven “leaky gut”	↑ Semen antioxidant capacity and motility ↑ Serum testosterone ↓ Inflammatory cytokines	Exercise–microbiome data [156,157,158,159,160] Stress–gut–testis data [161,162,163,164,165]
Antibiotic Stewardship	Prevents broad-spectrum antibiotic-induced dysbiosis that disrupts sperm or endocrine homeostasis	Maintenance of baseline sperm quality and hormonal balance	Case reports and epidemiological series [166,167,168,169,170]

Abbreviations: HPG, hypothalamic–pituitary–gonadal; MDA, malondialdehyde; PRDX6, peroxiredoxin-6; RCT, randomized controlled trial; SCFA, short-chain fatty acid; SOD, superoxide dismutase.

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
