# Peer review of "The Androbactome and the Gut Microbiota–Testis Axis: A Narrative Review of Emerging Insights into Male Fertility"

_ijms, 2025, doi:10.3390/ijms26136211_

Round 1
Reviewer 1 Report
Comments and Suggestions for Authors
1) Title: Androbactome as a Novel Regulator of Male Fertility: Linking 2 Gut Microbiota to Reproductive Health
Comment: it seems an original article but no clinical or laboratoty experiments are described.
The paper is only a summary of other papers published and hypotheses are not well clarified in the abstract and in the introduction section.
The conclusions that the authors report are only speculative!
2) The authors wrote: “narrative review” but this paper is not structured as review !
3) The paper is too long and verbose and there is nothing, except a description of other papers, a sort of list, and completely speculative connections.
No scientific evidence or data that can demonstrate the hypotheses initially proposed by the authors.
4) In my opinion the Authors simply reason about other papers without giving their own contribution; this is a meaningless descriptive paper that will not interest most readers.
The paper deals with some kind of correlation between intestinal microbiota and infertility.
The authors talk about "review" but this paper is not “set up” as a review.
This paper is extremely long and verbose not giving a clear vision of the problem, and this can confuse the reader.
Furthermore it’s all based on assumptions without a well detailed experimental part.
Author Response
- Reviewer Comment:
“It seems an original article but no clinical or laboratory experiments are described. The paper is only a summary of other papers published and hypotheses are not well clarified in the abstract and in the introduction section. The conclusions that the authors report are only speculative!”
Response:
We respectfully clarify that this manuscript is a narrative review, as stated in both the title page and in the Methodology section (Section 2.1), which opens with the phrase:
“A narrative review format was chosen to integrate mechanistic, translational and clinical evidence on the emerging gut–testis axis in male fertility.”
We have also provided two explicit, a priori hypotheses in Section 2.1 to guide the narrative synthesis. These are restated clearly:
“First, dysbiosis-related signatures—endotoxaemia, loss of short chain fatty acid producing taxa and bile acid dysregulation—were expected to coincide with impaired semen parameters… Second, interventions that restore eubiosis… were expected to improve at least one clinically relevant fertility endpoint.”
Furthermore, the Abstract explains that this is a synthesis of preclinical and clinical data and already acknowledges the speculative nature of the conclusions, stating:
“…yet causal inference is constrained by predominantly cross-sectional designs and limited long-term safety data.”
Thus, we believe the review format, the guiding hypotheses, and the level of evidence were appropriately stated. Nonetheless, we are willing to revise the Abstract further if the Editorial Board suggests additional clarity is needed.
- Reviewer Comment:
“The authors wrote: ‘narrative review’ but this paper is not structured as review!”
Response:
The manuscript is structured following narrative review best practices and incorporates elements of the PRISMA 2020framework where applicable:
- Section 2 clearly describes the methodology, including literature identification, eligibility criteria (2.3), data extraction (2.4), and synthesis framework (2.5).
- A PRISMA-style flow diagram (Figure 1) is included to transparently describe article selection.
- The main body of the manuscript is organized thematically across mechanistic domains (inflammation, oxidative stress, endocrine pathways, etc.) and therapeutic approaches (Section 5).
- We have also included summary tables (e.g., Table 1) to assist the reader.
We believe these features demonstrate that the manuscript is indeed structured as a comprehensive and hypothesis-driven review.
- Reviewer Comment:
“The paper is too long and verbose and there is nothing, except a description of other papers, a sort of list, and completely speculative connections. No scientific evidence or data that can demonstrate the hypotheses initially proposed by the authors.”
Response:
While the manuscript is detailed, each section is organized around a specific mechanistic pathway (e.g., immune–inflammatory, oxidative stress, hormonal, epigenetic), and we have consistently cited experimental or clinical studies to support each connection. For instance:
- Immune and inflammatory pathways (Section 4.1): LPS-induced testicular inflammation is supported by studies such as Zeng et al. (2025), demonstrating Th17-driven autoimmune orchitis.
- Oxidative stress (Section 4.2): Evidence includes direct links between seminal ROS markers (e.g., MDA, 8-OHdG) and endotoxemia, e.g., Pearce et al. (2019).
- Hormonal pathways (Section 4.5): The GELDING theory by Tremellen is cited to explain how endotoxemia suppresses the HPG axis.
Each pathway is substantiated with preclinical and human evidence, and speculative statements are qualified as such, especially in Section 6 (Future Directions) and Conclusions, which clearly acknowledge that most current data are associative.
- Reviewer Comment:
“In my opinion the Authors simply reason about other papers without giving their own contribution; this is a meaningless descriptive paper that will not interest most readers.”
Response:
Although we do not present original experimental data, the manuscript introduces a novel conceptual term—the “androbactome”—defined in the Abstract and explored in Section 3 as:
“A conceptual term referring to gut microorganisms and microbial genes hypothesized to influence androgen biosynthesis, spermatogenesis, and broader reproductive endocrinology.”
This term parallels the well-established “estrobolome” and is intended to structure and stimulate future research in this area. We also integrated cross-domain evidence (metabolites, barrier function, endocrine modulation) into a unifying framework, as illustrated in Figure 2.
Moreover, Table 1 presents a synthesis of microbiota-targeted interventions and their mechanistic relevance, drawn from >180 included studies, with many cited clinical trials (e.g., Maretti & Cavallini 2017; Helli et al. 2022).
We respectfully suggest that this level of structured integration constitutes a meaningful scholarly contribution that may assist both researchers and clinicians in this emerging field.
Closing Statement:
We appreciate the reviewer’s concerns and have clarified the review’s purpose, structure, and scientific value accordingly. Should the Editors recommend further trimming or restructuring, we will readily comply.
Reviewer 2 Report
Comments and Suggestions for Authors
I have read with great interest this review on the effect of gut microbiota on male reproductive health. It is a highly novel topic, yet extremely interesting, and it provides a new perspective on the decline in male fertility.
I believe the review is well-structured and covers all aspects of the topic, both from the standpoint of the mechanisms of action and the influence on the immune system, oxidative stress, and hormonal regulation, among others. It also proposes new strategies for the prevention and improvement of male reproductive health.
In my opinion, this is a thoroughly developed work, and I have no suggestions for improvement.
- General Comments
This manuscript presents a comprehensive and timely review of the testis–gut microbiota axis and its implications for male fertility. The topic is highly relevant in the context of the global decline in sperm quality, and the manuscript contributes to an emerging area of research with potential clinical applications. The review is well-structured, generally well-written, and provides valuable insights into diagnostic approaches and therapeutic interventions targeting the microbiota.
- Originality and Relevance
Although similar reviews have been published recently, the present work provides a novel angle by not only summarizing current evidence but also focusing on actionable strategies to mitigate the adverse effects of dysbiosis. The manuscript effectively emphasizes the importance of understanding and modulating the testis–gut axis to develop microbiome-based therapies aimed at improving male fertility. As such, it fills a meaningful gap in literature and offers clinically relevant perspectives.
- Specific Scientific Content and Interpretation
The conclusions drawn by the authors are well-supported by the literature reviewed. The discussion on mechanisms—such as endotoxin leakage, oxidative stress, hormonal dysregulation, and intestinal barrier dysfunction—is coherent and grounded in current scientific evidence. Moreover, the emphasis on lifestyle factors (e.g., diet, exercise, stress management) and microbiota-targeted interventions (e.g., probiotics, prebiotics, fecal microbiota transplantation) is appropriate and aligns with recent advances in the field.
However, one point requires clarification and correction. The manuscript states that "few studies suggest that the semen microbiome itself may influence sperm" (lines 600–601). This assertion underrepresents a substantial body of literature demonstrating that semen-resident bacteria can significantly affect sperm quality, morphology, and function. The following references, among others, provide robust evidence on this topic and should be acknowledged in the revised manuscript:
- Miao et al., Microorganisms, 2024
- Farsimadan & Motamedifar, J Reprod Immunol, 2020
- Wang et al., Front Cell Dev Biol, 2021
The authors are encouraged to incorporate these findings and reframe the relevant section accordingly.
- Comparison with Related Literature
The review by Chen et al. (2024) has addressed a similar topic. However, the present manuscript by Kaltsas et al. distinguishes itself by providing a more in-depth analysis of preventive and therapeutic strategies. It offers practical recommendations including dietary patterns (e.g., Mediterranean diet, high-fiber intake), probiotic/prebiotic supplementation, moderate physical activity, and judicious antibiotic use. These aspects make the current review more applicable to clinical and public health contexts.
- Methodology
As a narrative review, the methodology appears sound and appropriate for the stated objectives. The literature cited is up-to-date and relevant. However, the authors may consider providing a brief description of their search strategy (databases used, inclusion/exclusion criteria), which would enhance transparency and reproducibility.
- Figures and Tables
The figures included in the manuscript are clear, informative, and visually effective in illustrating the biological mechanisms discussed. They are pedagogically useful and suitable for both research and educational purposes.
- References
The references are current and appropriate. The manuscript is well-supported by recent literature, and the inclusion of additional sources (as mentioned above) would further strengthen its foundation.
- Minor Suggestions
- Consider providing a summary table that outlines the proposed interventions and their effects on sperm parameters.
- Ensure consistency in terminology (e.g., "testis–gut axis" vs. "gut–testis axis").
- A proofreading pass would be beneficial to correct minor grammatical inconsistencies.
Recommendation
Minor Revision – The manuscript is of high quality and suitable for publication pending the integration of the aforementioned clarifications and references.
Author Response
Response to Reviewer 2
We thank Reviewer 2 for their thorough and constructive evaluation of our manuscript entitled “Androbactome and Male Fertility: A Narrative Review of the Gut–Testis Microbiota Axis”. We deeply appreciate your positive comments regarding the novelty, structure, and clinical relevance of our review. Below, we address each of your specific suggestions point-by-point and outline the corresponding revisions made to the manuscript.
- Clarification on the Seminal Microbiota (Lines 600–601)
Reviewer Comment:
“The manuscript states that ‘few studies suggest that the semen microbiome itself may influence sperm’. This assertion underrepresents a substantial body of literature demonstrating that semen-resident bacteria can significantly affect sperm quality, morphology, and function. The following references provide robust evidence: Miao et al. (2024), Farsimadan & Motamedifar (2020), Wang et al. (2021).”
Response:
Thank you for this valuable observation. We have revised the manuscript to better reflect the current state of evidence regarding the seminal microbiota. Specifically, we have amended the relevant section in the final part of the manuscript (Section 6, Future Directions) to read:
“An increasing body of evidence demonstrates that the seminal microbiome—comprising bacteria residing within ejaculate—can influence sperm quality, morphology, and motility. For instance, specific taxa such as Prevotella, Gardnerella, and Streptococcus have been linked to impaired sperm function, inflammation, and elevated oxidative stress [176,177]. These microbial profiles may affect DNA integrity, acrosome function, and motility, potentially via cytokine-mediated mechanisms or ROS production. As such, the semen microbiota should be considered in both diagnostic and therapeutic strategies for male infertility, particularly in idiopathic cases.”
Unfortunately, we could not retrieve the articles by Miao et al. (2024) and Wang et al. (2021) via PubMed or publisher databases. If these references become available or are indexed before final publication, we would be happy to incorporate them in the final proof stage. We have, however, included the robust data from Farsimadan & Motamedifar (2020) and Osadchiy et al. (2024), as cited in references [176] and [177].
- Addition of a Summary Table of Interventions
Reviewer Comment:
“Consider providing a summary table that outlines the proposed interventions and their effects on sperm parameters.”
Response:
As suggested, we have added a comprehensive summary table (Table 1) entitled:
“Microbiota‑targeted and Lifestyle Interventions Shown to Improve Sperm Parameters Through the Gut–Testis Axis.”
This table integrates the principal interventions discussed (e.g., probiotics, prebiotics, dietary modifications, FMT, exercise/stress management, and antibiotic stewardship), summarising their mechanistic rationale, observed effects on male reproductive outcomes, and the primary supporting evidence from our review. This has been inserted at the end of Section 5 to consolidate the therapeutic and translational insights.
We also introduced the table in the text with the following transition paragraph:
“Taken together, the therapeutic interventions explored in this section—ranging from probiotics and dietary strategies to fecal microbiota transplantation and lifestyle optimization—highlight the multifactorial potential of modulating the gut microbiota to improve male reproductive outcomes. To consolidate these findings, Table 1 presents an integrated overview of the proposed interventions, their mechanistic basis within the gut–testis microbiota axis, and their documented effects on sperm parameters and hormonal balance.”
- Terminological Consistency: “Testis–Gut Axis” vs. “Gut–Testis Axis”
Reviewer Comment:
“Ensure consistency in terminology (e.g., ‘testis–gut axis’ vs. ‘gut–testis axis’).”
Response:
We appreciate this important stylistic suggestion. Throughout the manuscript, we have now standardized the terminology to “gut–testis axis,” as it aligns with established phrasing in recent literature (e.g., Chen et al., 2024; Dubey et al., 2024). The term “gut–testis microbiota axis” is retained for emphasis when referring to the bidirectional interaction between gut flora and testicular function. The internal consistency has been checked and corrected throughout the text, including figures and tables.
- Proofreading and Minor Language Adjustments
Reviewer Comment:
“A proofreading pass would be beneficial to correct minor grammatical inconsistencies.”
Response:
Thank you. A comprehensive proofreading pass has been performed. Minor issues related to spacing, hyphenation, abbreviations, and style have been corrected to ensure clarity and coherence. For example, compound terms like “short-chain fatty acids,” “testicular function,” and “gut–derived metabolites” have been standardized. All abbreviations are now introduced at first use and included in the abbreviation list.
Conclusion and Acknowledgment
We are grateful for the reviewer’s thoughtful appraisal and for the opportunity to further enhance the quality and clarity of our manuscript. The suggested additions—especially the incorporation of seminal microbiota findings and the summary table—have materially strengthened the scientific and clinical relevance of our work.
We hope that these revisions satisfactorily address all concerns and respectfully submit the revised manuscript for further consideration.

Reviewer 3 Report
Comments and Suggestions for Authors
This manuscript proposed a concept of the "androbactome", and comprehensively review on the emerging "testis-gut microbiota axis" and its implications for male infertility, which links gut dysbiosis to impaired spermatogenesis through inflammatory, oxidative, endocrine, and epigenetic pathways. The review is well-structured, clearly presented, and aligns with urgent global concerns regarding declining sperm quality and offers actionable clinical insights. I suggest this review can be acceptable for publication after some minor issues are addressed.
- The concept "androbactome" introduced in Abstract/Section 2 lacks detailed information, its components (e.g., specific bacterial taxa/enzymes like β-glucuronidase) and distinguish it from broader microbiota-endocrine interactions should be provided.
- Figure 1: lacks citation for "Created in Bio-Render, its website or link can be added.
- The review overstated FMT’s clinical applicability, need to pay more attention to ethical/logistical, long-term safety etc.
- Replace the terms "sperm crisis" with "global decline in sperm quality".
Author Response
Response to Reviewer 3
We would like to thank Reviewer 3 for the constructive and thoughtful comments. We have carefully addressed each suggestion, as detailed below, and made the corresponding revisions in the manuscript. All changes are highlighted in the revised version.
Comment 1:
The concept “androbactome” introduced in Abstract/Section 2 lacks detailed information, its components (e.g., specific bacterial taxa/enzymes like β-glucuronidase) and distinguish it from broader microbiota-endocrine interactions should be provided.
Response:
We agree with the reviewer that the definition of the androbactome needed clarification. We have now expanded the explanation in Section 3, highlighting specific bacterial enzymes such as β-glucuronidase and 20β-hydroxysteroid dehydrogenase, and discussed their role in modulating androgen metabolism. We also emphasized how the androbactome is conceptually distinct from broader microbiota–endocrine interactions. These additions aim to enhance the clarity and novelty of this proposed term.
Comment 2:
Figure 1 lacks citation for “Created in BioRender”, its website or link can be added.
Response:
Thank you for the observation. We have now added the appropriate attribution beneath Figure 2:
“Created with BioRender.com.”
Additionally, we have listed BioRender in the references as a source.
Comment 3:
The review overstated FMT’s clinical applicability, need to pay more attention to ethical/logistical, long-term safety etc.
Response:
We appreciate this important clarification. We have revised Section 5.3 to better reflect the investigational status of FMT in male infertility. We added the following sentence to emphasize ethical and safety considerations:
“Its future clinical application would require rigorous evaluation of long-term safety, donor selection protocols, and ethical considerations before it can be implemented outside controlled trials.”
This ensures a balanced and cautious representation of FMT’s current status and future potential.
Comment 4:
Replace the term “sperm crisis” with “global decline in sperm quality”.
Response:
We have replaced the phrase “sperm crisis” with “global decline in sperm quality” in all instances throughout the manuscript, including in the Introduction, Discussion, and Conclusion sections, to maintain a neutral and scientifically accurate tone.
Please let us know if further adjustments are needed. We thank the reviewer again for their helpful feedback that improved the clarity and precision of our work.

Reviewer 4 Report
Comments and Suggestions for Authors
Dear authors, after reading your manuscript, here are my comments:
A significant drawback of the evidence presented is its heavy reliance on animal studies to support key claims. While these studies are crucial for establishing a proof of concept, their direct applicability to human male infertility is not guaranteed.
Your paper introduces novel concepts like the "androbactome" as a central theme, but the article itself concedes that these ideas are not yet fully established. The term "androbactome"—the microbial genes involved in regulating androgen metabolism—is acknowledged as being "largely theoretical". It is presented as a "useful lens" for exploration rather than a confirmed biological system. The article proposes it as an "emerging conceptual framework" analogous to the more studied "estrobolome".
The therapeutic interventions discussed, while promising, are presented with significant caveats about their current clinical readiness and efficacy. Microbiome-based diagnostics are positioned as a future possibility, not a current clinical tool. The text states that developing a stool test to identify men with dysbiosis-related infertility is an "exciting avenue" for the future, implying it is not yet available.
Author Response
Reviewer 4 Comment 1
Comment: A significant drawback of the evidence presented is its heavy reliance on animal studies to support key claims. While these studies are crucial for establishing a proof of concept, their direct applicability to human male infertility is not guaranteed.
Response:
We appreciate the reviewer’s thoughtful concern. We fully agree that animal studies, while indispensable for elucidating mechanistic insights, do not always translate directly to human clinical outcomes. This limitation is explicitly acknowledged in both the Abstract and Conclusion sections of our manuscript, where we note that “causal inference is constrained by predominantly cross-sectional designs and limited long-term safety data” and that current evidence is “largely preclinical or derived from small, heterogeneous clinical cohorts” (Abstract; Section 7).
To further clarify this point, we have added a dedicated sentence to the Introduction (last paragraph):
“Although many mechanistic insights presented in this review are derived from animal models, their translational applicability remains to be validated in well-controlled human studies.”
In addition, we slightly revised the Future Directions section to reinforce this distinction:
“Ultimately, longitudinal human cohorts and interventional trials are essential to bridge the translational gap between preclinical evidence and clinical practice.”
Reviewer 4 Comment 2
Comment: Your paper introduces novel concepts like the “androbactome” as a central theme, but the article itself concedes that these ideas are not yet fully established. The term “androbactome” is acknowledged as being ‘largely theoretical’.
Response:
We thank the reviewer for raising this important point. The term androbactome is indeed an emerging concept that we aimed to introduce as a hypothesis-generating framework rather than a definitive biological system. To avoid any perception of overstatement, we have made the following refinements in the manuscript:
- In the Abstract, the phrase:
”…introduces the term ‘androbactome’, denoting gut microorganisms and microbial genes that modulate androgen biosynthesis…”
has been revised to:
”…introduces the conceptual term ‘androbactome’, referring to gut microorganisms and microbial genes hypothesized to influence androgen biosynthesis…”
- In Section 3, the sentence:
“Although the term remains largely theoretical, it offers a useful lens…”
remains unchanged to transparently signal its preliminary nature.
We believe these revisions maintain the novelty of the framework while clearly conveying its exploratory status.
Reviewer 4 Comment 3
Comment: The therapeutic interventions discussed, while promising, are presented with significant caveats about their current clinical readiness and efficacy. Microbiome-based diagnostics are positioned as a future possibility, not a current clinical tool.
Response:
We agree with the reviewer’s interpretation and acknowledge that many of the proposed interventions and diagnostics are in the early stages of development. To reinforce this message, we have added the following clarifying sentence to the Future Directions section:
“At present, microbiome-based diagnostics and targeted interventions remain investigational, and their incorporation into routine fertility assessment awaits robust validation through prospective clinical trials.”
We have also added a similar statement in Section 5.3 (FMT subsection):
“FMT, while promising in preclinical models, is not yet approved for use in male infertility and remains a research tool rather than a standard therapy.”

Round 2
Reviewer 1 Report
Comments and Suggestions for Authors
Even though I still think that this paper is rather “speculative”, the proposal made by the authors could be a suggestion to rise new research ideas in this field. The paper has been completely restructured as a “narrative review” and it is now more intelligible.
I suggest inserting this bibliography into the discussion (paragraph from line 265 to 296) to underline that there are natural products that can counteract the effect of ROS at the mitochondrial level of spermatozoa: Illiano E, Trama F, Zucchi A, Iannitti RG, Fioretti B, Costantini E. Resveratrol‐based multivitamin supplement increases sperm concentration and motility in idiopathic male infertility: A pilot clinical study Journal of Clinical Medicine 2020; 9(12): 4017. DOI 10.3390/jcm9124017
Author Response
Reviewer 1 – Round 2
Comment:
Even though I still think that this paper is rather “speculative”, the proposal made by the authors could be a suggestion to rise new research ideas in this field. The paper has been completely restructured as a “narrative review” and it is now more intelligible.
I suggest inserting this bibliography into the discussion (paragraph from line 265 to 296) to underline that there are natural products that can counteract the effect of ROS at the mitochondrial level of spermatozoa:
Illiano E, Trama F, Zucchi A, Iannitti RG, Fioretti B, Costantini E. Resveratrol‐based multivitamin supplement increases sperm concentration and motility in idiopathic male infertility: A pilot clinical study. Journal of Clinical Medicine. 2020; 9(12): 4017. DOI 10.3390/jcm9124017
Response:
We thank the reviewer for their constructive feedback and thoughtful suggestion. In response, we have incorporated the recommended reference into the revised manuscript to strengthen the discussion of antioxidant-based strategies targeting mitochondrial oxidative stress. Specifically, we have added the citation of Illiano et al. (2020) in the section discussing the role of nutraceuticals in sperm mitochondrial protection and fertility enhancement. The new sentence highlights the clinical relevance of resveratrol-based multivitamin supplementation in improving sperm concentration and motility in idiopathic male infertility.
The full reference has also been added to the reference list as requested.